# Bone Mineral Density and Trabecular Bone Score Changes throughout Menopause in Women with HIV

**DOI:** 10.3390/v15122375

**Published:** 2023-12-01

**Authors:** Jovana Milic, Stefano Renzetti, Denise Morini, Federico Motta, Federica Carli, Marianna Menozzi, Gianluca Cuomo, Giuseppe Mancini, Mattia Simion, Federico Romani, Anna Spadoni, Irene Baldisserotto, Nicole Barp, Chiara Diazzi, Chiara Mussi, Cristina Mussini, Vincenzo Rochira, Stefano Calza, Giovanni Guaraldi

**Affiliations:** 1Modena HIV Metabolic Clinic, University of Modena and Reggio Emilia, 41121 Modena, Italy; jovana.milic@gmail.com; 2Department of Surgical, Medical, Dental and Morphological Sciences, University of Modena and Reggio Emilia, 41121 Modena, Italy; federico.motta@unimore.it (F.M.); cristina.mussini@unimore.it (C.M.); 3Department of Medical-Surgical Specialties, Radiological Sciences and Public Health, University of Brescia, 25121 Brescia, Italy; stefano.renzetti@unibs.it; 4Hematology and Transplant Center, University Hospital “San Giovanni di Dio e Ruggi d’Aragona”, 84121 Salerno, Italy; deniseiriomorini@gmail.com; 5Department of Infectious Diseases, Azienda Ospedaliero-Universitaria, Policlinico of Modena, 41121 Modena, Italy; fedecarli@gmail.com (F.C.); marymenozzi@gmail.com (M.M.); gian.cuomo@gmail.com (G.C.); giusmancini3@gmail.com (G.M.); mattiasimion94@gmail.com (M.S.); romanif240@gmail.com (F.R.); anna.spadoni@icloud.com (A.S.); irenebaldisserotto@gmail.com (I.B.); nicole.barp94@gmail.com (N.B.); 6Unit of Endocrinology, Department of Biomedical, Metabolic and Neural Sciences, University of Modena and Reggio Emilia, 41126 Modena, Italy; chiaradiazzi@gmail.com (C.D.); vincenzo.rochira@unimore.it (V.R.); 7Department of Biomedical and Metabolic Sciences and Neuroscience, University of Modena and Reggio Emilia, 41126 Modena, Italy; chiara.mussi@unimore.it; 8Unit of Endocrinology, Department of Medical Specialties, Azienda Ospedaliero-Universitaria of Modena, Ospedale Civile of Baggiovara, 41126 Modena, Italy; 9Department of Molecular and Translational Medicine, University of Brescia, 25121 Brescia, Italy; stefano.calza@unibs.it

**Keywords:** bone mineral density, trabecular bone score, menopause, women with HIV

## Abstract

Objective: The objectives of this study were to describe the trajectories of bone mineral density (BMD) and trabecular bone score (TBS) changes throughout pre-menopause (reproductive phase and menopausal transition) and post-menopause (early and late menopause) in women with HIV (WWH) undergoing different antiretroviral therapies (ARTs) and explore the risk factors associated with those changes. Methods: This was an observational longitudinal retrospective study in WWH with a minimum of two DEXA evaluations comprising BMD and TBS measurements, both in the pre-menopausal and post-menopausal periods. Menopause was determined according to the STRAW+10 criteria, comprising four periods: the reproductive period, menopausal transition, and early- and late-menopausal periods. Mixed-effects models were fitted to estimate the trajectories of the two outcomes (BMD and TBS) over time. Annualized lumbar BMD and TBS absolute and percentage changes were calculated in each STRAW+10 time window. A backward elimination procedure was applied to obtain the final model, including the predictors that affected the trajectories of BMD or TBS over time. Results: A total of 202 WWH, all Caucasian, were included. In detail, 1954 BMD and 195 TBS data were analyzed. The median number of DEXA evaluations per woman was 10 (IQR: 7, 12). The median observation periods per patient were 12.0 years (IQR = 8.9–14.4) for BMD and 6.0 years (IQR: 4.3, 7.9) for TBS. The prevalence of osteopenia (63% vs. 76%; *p* < 0.001) and osteoporosis (16% vs. 36%; *p* < 0.001) increased significantly between the pre-menopausal and post-menopausal periods. Both BMD (1.03 (±0.14) vs. 0.92 (±0.12) g/cm^2^; *p* < 0.001) and TBS (1.41 (IQR: 1.35, 1.45) vs. 1.32 (IQR: 1.28, 1.39); *p* < 0.001) decreased significantly between the two periods. The trend in BMD decreased across the four STRAW+10 periods, with a slight attenuation only in the late-menopausal period when compared with the other intervals. The TBS slope did not significantly change throughout menopause. The delta mean values of TBS in WWH were lower between the menopausal transition and reproductive period compared with the difference between menopause and menopausal transition. Conclusions: Both BMD and TBS significantly decreased over time. The slope of the change in BMD and TBS significantly decreased in the menopausal transition, suggesting that this period should be considered by clinicians as a key time during which to assess bone health and modifiable risk factors in WWH.

## 1. Introduction

Menopause is a physiological hallmark of aging and predicts comorbidities, in particular, osteoporosis, in women better than chronological age. To facilitate research on menopause, the Staging of Reproductive Aging Workshop (STRAW) recommendations depict a continuum of events from the reproductive to the transitional and menopausal periods based on variations in menstrual cycle characteristics, hormone levels, antral follicle count, and symptoms in each stage [1,2,3].

The onset of menopause is determined by multiple factors, including, among others, genetic heritage, lifestyle (especially smoking), and socio-economic status. In women with HIV (WWH), the virus itself and possibly antiretroviral therapy (ART) are among the risk factors for menopause, both certainly associated with an increased risk of comorbidities, in particular, osteoporosis. The prevalence of osteoporosis in post-menopausal WWH is 15%, with a 58% higher fracture rate compared with the general population [4,5,6].

This greater disease burden has sometimes been attributed to the impact of ARTs on bone metabolism. Multiple studies, both on HIV and outside an HIV setting, found that tenofovir disoproxil fumarate (TDF)-containing ART regimens, but not tenofovir alafenamide (TAF), an alanine ester prodrug of TDF, led to greater bone loss, observed both in the spine and femur neck [5,7,8,9,10,11,12,13,14]. Protease inhibitors (PIs) are also associated with greater bone loss, particularly in the lumbar spine, and may also increase fracture risk [1,5,8,15,16,17,18]. However, other studies did not show the impact of TDF or other drug types on bone loss in people with HIV [19,20]. Moreover, ART-induced bone loss may be mediated by inflammatory responses related to CD4 rebound in severely compromised people with HIV [21].

The clinical picture of osteoporosis in menopause is portrayed by falls and fragility fractures leading to increased mortality, morbidity, hospitalization, and placement in nursing homes [5,22].

Screening for osteoporosis in WWH is recommended using the same criteria as in the general population, including post-menopause, history of low-impact fracture, and high fracture risk [23]. Prevention strategies start with fracture-risk stratification, which is assessed using Dual-Energy X-ray Absorptiometry (DEXA); this method, using dedicated software, measures bone quantity, according to bone mineral density (BMD), and bone quality, according to the trabecular bone score (TBS), which are seldom assessed in post-menopausal WWH.

The trabecular bone score (TBS) evaluates the trabecular microarchitecture derived from spine DEXA images in a quantifiable manner. The TBS can enhance fracture prediction when used in conjunction with BMD and clinical risk factors. The TBS offers additional data that may change the FRAX prediction [24].

The TBS has been studied in older individuals and found to independently predict hip and significant osteoporotic fractures even without considering BMD and other clinical risk factors. However, studies in people with HIV, in particular, WWH, are limited [25]. We hypothesized that BMD and TBS trends throughout menopause might be different in WWH and might be associated with both traditional and HIV-related risk factors.

The objectives of this study were to describe the trajectories of bone quantity (BMD) and bone quality (TBS) changes throughout pre-menopause (reproductive phase and menopausal transition) and post-menopause (early and late menopause) in WWH undergoing different ARTs and explore the risk factors associated with these changes.

## 2. Material and Methods

### 2.1. Study Design and Subjects

We conducted an observational longitudinal retrospective study in WWH attending Modena HIV Metabolic Clinic (MHMC) from January 2012 to December 2022. In this multidisciplinary tertiary care center, people with HIV (PWH) are screened for comorbidities and geriatric syndromes. Patients receive free-of-charge routine endocrinological screening, including hypothalamic–pituitary–gonadal function, as well as evaluation of bone metabolism and total body DEXA. Clinical activities were interrupted from February to October 2020 during the first wave of the COVID-19 epidemic in Italy.

Eligible subjects were WWH, ART-experienced for at least 12 months, older than 40 years, with an evaluable estimated date of menopause (as described below) and records of biochemical parameters assessing bone metabolism. A minimum of two DEXA evaluations comprising BMD and TBS measurements, both in the pre-menopausal and post-menopausal periods, were requested per protocol for each patient.

Women were excluded if, at the time of the first DEXA evaluation, they were already post-menopausal or had co-morbidities affecting bone turnover metabolism such as multiple myeloma, cancer, endocrinopathies, serum creatinine above 1.5 mg/dL, celiac or inflammatory bowel disease, current glucocorticoid or anticonvulsant use, current hormone replacement therapy, and current or past treatment of osteoporosis with bone-active drugs.

Among 560 WWH attending MHMC, 202 (36.1%) were post-menopausal at the end of the observational period. The main criterion for menopause classification according to STRAW+10 is the absence of menstrual bleeding for more than 1 year or a history of total abdominal hysterectomy with bilateral salpingo-oophorectomy. In addition, supportive criteria included biomarkers of ovary insufficiency using the cutoffs of luteinizing hormone (LH) > 25 ng/mL, follicle-stimulating hormone (FSH) > 25 ng/mL, and estradiol (E2) < 30 ng/mL [26].

The date of menopause was arbitrarily estimated as the mean time between two consecutive endocrinological assessments presenting a fertile gonadal status at baseline and ovary insufficiency at follow-up. For hysterectomized women, the time since menopause was estimated as years since the onset of menopausal symptoms.

In each woman, the observation period, and consequently, the time in which DEXA assessments were performed, was divided into four intervals according to the 2011 STRAW+10 classification [26] (Appendix A) as follows.

“Reproductive Period” (T1): up to 2 years before menopausal transition.“Menopause Transition Period” (T2): from 2 years before menopause to menopause.“Early Menopause Period” (T3): from menopause onset to 6 years post-menopause.“Late Menopause Period” (T4): from 6 years post-menopause onwards.

Due to the relatively limited number of women contributing to each STRAW+10 period, descriptive data are presented in reference to the pre- and post-menopausal periods.

### 2.2. Covariates and Outcomes

We recorded data indicative of lifestyle; medical, surgical, and gynecological history; osteoporosis risk factors; current and past medication history; and HIV and ART history. HIV-related variables included current CD4 cell count and nadir CD4 cell count, years since HIV diagnosis, and percentage of PWH with undetectable HIV RNA viral load (<50 copies/mL). ART history was obtained using an interview and detailed chart review and included current exposure to TDF, TAF, and protease inhibitors. TAF was introduced into the Italian market in 2016, and second-generation integrase inhibitors (INSTIs) were introduced in 2015.

Smoking was assessed as current smoking status. As part of the standard of care in the clinic, vitamin D (VitD) levels were monitored at each clinical visit, and vitamin D supplementation was prescribed to those participants with baseline levels <30 ng/mL. A standard 10,000 IU VitD weekly dose was prescribed according to European AIDS Clinical Society (EACS) guidelines [27]. Treatment with bisphosphonates in pre- and post-menopause was included in the models. A few WWH used alternative osteoporosis treatment with teriparatide or denosumab and were thus excluded.

Endocrinological variables included LH, FSH, and E2 and were measured at each visit. LH and FSH were assayed with a chemiluminescent immunoassay, while E2 was measured using two validated isotopic dilution-liquid chromatography–tandem mass spectrometry assessments [28].

### 2.3. Bone Quantity and Bone Quality Assessments

The BMD values of the lumbar spine (L-BMD; L1–4) and femoral neck (F-BMD) were measured using DEXA (Hologic, Inc., Technologic srl, Rome, Italy). DEXA underwent calibration daily to ensure accurate BMD measurements using an anthropometric phantom. Osteoporosis and osteopenia were defined according to World Health Organization (WHO) criteria for post-menopausal Caucasian women: T-scores of −2.5 or less represented osteoporosis, and T-scores between −1.0 and −2.49 represented osteopenia [29].

The trabecular bone score (TBS) was measured using DEXA with the same densitometer using TBS InSight software version 3.0 (Medimaps^®^). This software has been available at MHMC since 2019. TBS is a recently introduced metric that analyzes gray-level textual information in lumbar spine DEXA images to approximate the trabecular microarchitecture. TBS is calculated using an extrapolation technique from a low-resolution 2D image, and it is not impacted by bone size.

TBS values increase with enhanced skeletal texture (i.e., better microarchitecture) and decrease with deteriorated skeletal texture (i.e., poorer microarchitecture). Numerous studies have found a correlation between TBS texture parameters and microstructural parameters of bone as determined using micro-CT (exclusively preclinical technology and not suitable for humans), indicating a relationship between TBS and 3D microarchitecture [30,31].

Fracture risk was assessed using the FRAX^®^ score (version 4.2) [27]. The country-specific (Italy) 10-year probability of a major fracture (spine, wrist, proximal humerus, and hip) was calculated by adding femoral neck BMD assessment to the fracture-risk algorithm and including HIV as a secondary cause of osteoporosis [32,33].

This was a retrospective study conducted using clinical data anonymized in accordance with the requirements of the Italian Personal Data Protection Act. Patients’ consent was deemed unnecessary by the Regional Ethics Committee of Emilia Romagna according to Italy’s Legislative Decree No. 211/2003. The study was conducted according to the guidelines of the Declaration of Helsinki.

### 2.4. Statistical Analysis

All continuous variables were summarized as means and standard deviations if they showed a symmetric distribution, or medians, and 1st and 3rd quartiles in case of an asymmetric distribution. The Shapiro–Wilk test was used to estimate the normality of distribution. Counts and frequencies were used for categorical variables. The paired *t*-test or paired Wilcoxon rank-sum test was used for normally or non-normally distributed continuous variables, respectively, while the McNemar test was applied to categorical variables. Two linear mixed-effects models were fitted to estimate the trajectories of the two outcomes (BMD and TBS) over time. Annualized lumbar BMD and TBS absolute and percentage changes were calculated in each STRAW+10 time window. A backward elimination procedure was applied to obtain the final model, including the predictors that affected the trajectory of BMD or TBS over time. With a regression model, we estimated a three-way interaction term including (i) each covariate, (ii) time, and (iii) variables in the 4 STRAW+10 windows. For both pre- and post-menopause, the initial full model included the following covariates: use of bisphosphonates, TDF or PI, chronic kidney disease (<60 mil/min), nadir CD4 l < 200/mmc, current CD4 > 500/mmc, smoking status, moderate/hazardous alcohol use, and overweight (BMI > 25). All models were also adjusted for age at the first visit.

All tests were two-sided, and the statistical significance was set to 5% (*p* < 0.05). Statistical open-source R software (version 4.2.2) was used to analyze the data.

## 3. Results

### 3.1. Study Population

A total of 202 WWH, all Caucasian, were included in the observational period, contributing to 1978 DEXA assessments. In detail, 1954 BMD and 195 TBS data points were analyzed. The median number of DEXA evaluations per woman was 10 (IQR: 7, 12). The median observation periods per patient were 12.0 years (IQR = 8.9–14.4) for BMD and 6.0 years (IQR: 4.3, 7.9) for TBS.

According to the STRAW+10 classification, DEXA results were grouped as shown below.

“Reproductive Period”: 630 (32.2%) BMD and 40 (20.5%) TBS data points.“Menopause Transition Period”: 301 (15.4%) BMD and 49 (25.1%) TBS data points.“Early Menopause Period”: 878 (44.9%) BMD and 104 (53.3%) TBS data points.“Late Menopause Period”: 145 (7.4%) BMD and 2 (1.0%) TBS data points.

Table 1 summarizes the demographic and clinical characteristics of the study population in the pre- and post-menopausal periods according to BMD and in the subset of WWH with available TBS. In the pre-menopausal period, the median age was 44.2 (IQR: 41.8, 46.7) years, and the median BMI was 22.1 (IQR: 20.2, 24.3). The median CD4 cell count was 606 cells/µL (IQR: 433, 775); the median nadir CD4 cell count was 183 cells/µL (IQR: 100, 254); the mean time since HIV diagnosis was 18 (±6) years; and undetectable HIV RNA was achieved in 165 individuals (83%) (Table 1).

### 3.2. Changes throughout Menopause

The prevalence of osteopenia (63% vs. 76%; *p* < 0.001) and osteoporosis (16% vs. 36%; *p* < 0.001) increased significantly between the pre-menopausal and post-menopausal periods. Both BMD (1.03 (±0.14) vs. 0.92 (±0.12) g/cm^2^; *p* < 0.001) and TBS (1.41 (IQR: 1.35, 1.45) vs. 1.32 (IQR: 1.28, 1.39); *p* < 0.001) decreased significantly between the two periods (Table 1).

The use of bisphosphonates (16% vs. 3.5%; *p* < 0.001) and TAF (23% vs. 1%; *p* < 0.001) was higher, while the use of TDF (73% vs. 48%; *p* < 0.001) and boosted PIs (62% vs. 52%; *p* = 0.009) decreased in the post-menopausal period (Table 1).

The annualized BMD and TBS changes in the STRAW+10 time windows were −1.5% and −1.6% in the reproductive period, and −1.8% and −1.1% in the menopausal transition period, respectively. Further, there was a −1.7% change in the early-menopausal period and a −1.1% change in the late-menopausal period in BMD, while there was a change of −1.4% in TBS in the menopausal period.

Figure 1 depicts the trajectory of BMD throughout menopause. Panel A presents the slope coefficient of BMD change in the four STRAW+10 intervals. It can be observed that the trend in BMD decreased across the four STRAW+10 periods ($\beta_{1}$ = −0.016, *p* < 0.001; $\beta_{2}$ = −0.018, *p* < 0.001; $\beta_{3}$ = −0.016, *p* < 0.001; $\beta_{4}$ = −0.011, *p* < 0.001), with a slight attenuation only in the late-menopausal period when compared with the other intervals (T4–T1: 0.005, *p* = 0.017; T4–T2: 0.007, *p* = 0.035; T4–T3: 0.006, *p* = 0.014). Panel B describes the average values of BMD in each transition period. The delta mean values of BMD in WWH appear lower when considering the difference between the menopausal transition and the reproductive period compared with the differences relative to the following time points. The trajectories of the BMD Z-score throughout menopause according to the STRAW+10 intervals show that the BMD Z-scores were significantly lower during early menopause in WWH in comparison with the reference population ($\beta_{3}$ = −0.05, *p* < 0.001) (Appendix A).

Figure 2 depicts the trajectory of TBS throughout menopause using the same format as Figure 1. Due to the limited TBS data in the early- and late-menopausal periods, data were grouped in a single post-menopausal period. It can be observed that the TBS slope does not significantly change throughout menopause. The delta mean values of TBS in WWH appear lower between the menopausal transition and reproductive period compared with the difference between menopause and the menopausal transition.

Figure 3 includes the variables that were selected using the backward selection method and that showed an impact on BMD change throughout menopause. In detail, use of bisphosphonates (Figure 3A, Appendix A), kidney dysfunction (Figure 3B, Appendix A), BMI > 25 (Figure 3C, Appendix A), and use of PIs (Figure 3E, Appendix A) showed different patterns of BMD over time among groups, while exposure to TDF (Figure 3D, Appendix A) and CD4 > 500 c/microL (Figure 3F, Appendix A) did not.

WWH that were taking bisphosphonates showed a more horizontal trend starting from a lower level of BMD compared with those who were not taking bisphosphonates (bisphosphonates: yes vs. no = 0.138, *p* < 0.001) and a higher level of BMD in the late-menopausal stage (bisphosphonates: yes vs. no = 0.038, *p* = 0.013) (Figure 3A, Appendix A).

WWH with kidney dysfunction showed similar BMD levels to those with no dysfunction but a higher BMD level at the late-menopausal time point (kidney dysfunction: yes vs. no = 0.047, *p* = 0.002) (Figure 3B, Appendix A). WWH with overweight had higher BMD levels in all menopausal stages (Figure 3C, Appendix A).

WWH who were taking PIs showed lower BMD during the menopausal transition (PIs: yes vs. no = 0.045, *p* < 0.001) and early menopause (PIs: yes vs. no = −0.025, *p* = 0.001) (Figure 3E, Appendix A). CD4 > 500/microL and TDF use did not show significantly different trends in BMD over time, but the average difference in each STRAW+10 period was statistically significant, showing lower BMD levels among subjects with CD4 greater than 500 (CD4 > 500 vs. CD4 <= 500 = −0.017, *p* = 0.043; Figure 3F, Appendix A) or taking TDF (TDF: yes vs. no = −0.025, *p* < 0.001; Figure 3E, Appendix A).

Figure 4 includes the variables that were selected using the backward selection method and that showed an impact on TBS change throughout menopause. In detail, BMI > 25 (Figure 4C, Appendix A) and the use of PIs (Figure 4D, Appendix A) showed different patterns in TBS over time among groups. WWH with overweight (Figure 4C, Appendix A) showed a different pattern from WWH with normal weight, but the average TBS levels in each STRAW+10 period were similar to the ones of normal-weight patients.

WWH that were taking PIs showed lower TBS during the menopausal transition (PIs: yes vs. no = −0.045, *p* <0.001) and early menopause (PIs: yes vs. no = −0.041, *p* = 0.037; Figure 4D, Appendix A). Bisphosphonates and smoking status did not show significantly different trends in TBS over time, but the average difference in each STRAW+10 period was statistically significant, showing lower TBS levels among subjects taking bisphosphonates (bisphosphonates: yes vs. no = −0.073, *p* = 0.002; Figure 4A, Appendix A) or smokers (smoker: yes vs. no = −0.034, *p* = 0.041; Figure 4B, Appendix A).

## 4. Discussion

This study examined both bone quantity (using BMD) and bone quality (using TBS) trajectories in WWH across four distinct time windows, namely, the reproductive phase, transitional period, and early and late menopause. We hypothesized that classical and HIV-specific variables, including TDF or boosted PI-induced bone toxicity, may have a different impact throughout menopause according to the STRAW time periods.

With regard to bone disease, as expected, we observed an increase in the number of women with osteopenia and osteoporosis throughout menopause. Similarly, we observed an increase in the number of WWH with intermediate degraded microarchitecture throughout menopause. Likewise, Yin et al. showed that the prevalence of osteopenia, osteoporosis, and impaired TBS were higher in HIV-infected post-menopausal women [25,34,35].

In parallel, we observed a significant increase in the FRAX^®^ score throughout menopause (major osteoporotic fracture, from 3% to 4%); this is expected to be a function of age. Nevertheless, the mean fracture risk was still within the low fracture-risk probability. It should be noted that the FRAX^®^ score has not been specifically validated in the HIV population, and at a clinical level, bisphosphonates in WWH should not be considered exclusively when estimating fracture-risk probability with FRAX [22].

Our cohort was very well-characterized and assessed at multiple time points during a median 6-year follow-up. We were able to minimize the uncertainty in menopause onset by performing two consecutive endocrinological assessments within 2 years. The mean age at menopause was 49.3 years, similar to what is expected in the general population [36,37]. These data are apparently in conflict with what was reported by De Pommerol and Schoenbaum, who described the earlier onset of menopause in WWH, but these studies did not standardize the definition of menopause using endocrinological evaluations [38,39].

According to our study, we were able to describe osteoporosis as an age-related condition and as a continuum of events from the reproductive period to late menopause. We were able to show that both the absolute and mean values of BMD and TBS significantly decreased over time as a continuum highly dependent on endocrinological changes throughout menopause. The novelty of this study is the capturing of the reduction in BMD and TBS described by changes in the slope observed in the menopausal transition period and not only in the post-menopausal period. In the general population, the slope changes in a more dramatic way during early menopause compared with pre-menopause. Additionally, our findings are further reinforced by the fact that the trend in the BMD Z-score was significantly lower in the early-menopausal period compared with the reference population. Increased bone loss in the pre-menopausal period in WWH may be due to HIV itself and the use of antiretroviral drugs associated with bone toxicity.

This observation has important clinical implications, suggesting that bone health assessment and intervention need to be proactively initiated in the transition period rather than in the menopausal period. As an example, DEXA evaluation with possible anti-resorptive treatment should be considered in the pre-menopausal period and not only in the post-menopausal period.

We identified several risk factors associated with BMD and TBS. In our study, WWH with overweight had higher levels of BMD in both pre-menopausal and post-menopausal periods. This finding is usually justified by mechanical loading and the effects of hormones secreted or regulated by adipocytes [40]. However, the relationship between obesity and BMD remains unclear. Although some studies show that obesity may be a protective predictor of osteoporosis [41], other studies imply a differential impact of specific fat compartments on BMD [42], suggesting that BMI might not be a suitable marker of obesity. Our model did not find an association between TBS and BMI, similar to what was observed in the general population [43], while the association between fat compartments and TBS was not studied.

Smoking is a recognized risk factor for both early menopause and osteoporosis, and it is associated with decreased mineral density and TBS [44,45]. However, the association between smoking and bone quality throughout menopause is lacking. In our study, smokers had lower levels of TBS in all time periods, suggesting that smoking affects bone quality not only before the onset of menopause but also in post-menopause. It has been hypothesized that smoking may inhibit the vitamin D–PTH axis in post-menopausal women, in particular, in older women, by reducing the absorption of calcium [46].

As expected, WWH with a higher BMD score had lower probability of being exposed to bisphosphonates before menopause, but they also experienced progressive BMD loss throughout menopause. On the other hand, WWH using bisphosphonates did not present significant BMD loss between the menopausal transition and early menopause. This trend was not confirmed for TBS, implying a limited role of bisphosphonates in improving bone quality, even if initiated before menopause onset. However, the data in the general population regarding TBS and the use of bisphosphonates are conflicting. A small longitudinal study conducted in treated post-menopausal women showed that TBS increased over time (although to a lower extent than BMD) [47], while a recently published study showed the insensitivity of TBS to oral bisphosphonates; it is unlikely to be a clinically meaningful tool to evaluate treatment success, but it remains useful in fracture prediction [48].

Considering HIV-related variables, as expected, high CD4 cell counts, the use of TDF, and boosted PIs were identified as risk factors affecting BMD and TBS throughout menopause. As suggested previously, CD4 cell counts > 500 c/microL may be a proxy of increasing age, associated with age-related conditions [49], including osteoporosis and low BMD. This was similarly confirmed in this study. Also, this trend may be attributed to significant inflammatory effects associated with estrogen deficiency on CD4 cells [50,51].

The impact of TDF on bone health has been widely described, and it is associated with an increased risk of osteopenia/osteoporosis [14]. The additional value of our study is the ability to describe the trends in BMD changes throughout menopause in WWH who did or did not use TDF, in line with previous findings [52]. However, the use of TDF was not associated with decreasing trends in TBS, suggesting that bone toxicity induced by TDF does not involve the bone architecture. Another mechanism of bone toxicity related to antiretroviral drugs could be attributed to the use of boosted PIs. Interestingly, our study depicted the significant loss of both BMD and TBS in WWH taking PIs, implying that boosted regimens might display different mechanisms of bone toxicity from TDF. Moreover, it appears that BMD loss and TBS loss are more apparent in the menopausal transition and early menopause than in late menopause, suggesting that PIs should be avoided in these periods, if possible. Additionally, in this period, some WWH with troublesome vasomotor symptoms may take estrogen therapy, which has significant drug–drug interactions with PIs.

Overall, TBS appears to be a more conservative marker of bone health, as it is apparently not affected by estrogen-sensitive bone tissue. The architecture, accumulation of microdamage, turnover rate, mineralization, and the properties of the matrix are the parameters affecting bone quality. The potential of TBS as an adjunct to BMD for better prediction of fragility fractures in WWH has been explored in few studies. Because of the importance of bone quality factors in bone strength, additional assessment of the bone microarchitecture using TBS may better identify WWH at risk of fracture, as suggested by Sharma et al. [1]. For this reason, comparing BMD and TBS throughout menopause may be useful in the future to better identify WWH who are likely to experience more fragility fractures in menopause. However, in a study by Yang et al., the predictive accuracy of FRAX for hip fracture and major osteoporotic fracture did not further improve with the addition of TBS [53], contrary to what was described in the general population [48].

At present, the majority of WWH in industrialized countries are entering the menopausal transition. In this scenario, properly designed clinical studies are needed to address immediately effective interventions to reduce the burden of osteoporosis and fracture risk in the post-menopausal period.

Some strengths of this study must be acknowledged. Firstly, menopause was defined using the 2011 STRAW+10 staging system, which is widely considered the gold standard for characterizing reproductive aging through menopause. These criteria could serve as an excellent tool to assess ovarian aging in the wider context of the aging epidemic occurring in PWH. Secondly, we described a relatively large cohort of WWH. All the women were longitudinally observed for a median of 6 years of follow-up and a median of four DEXA assessments. This allowed us to obtain detailed data describing BMD and TBS changes throughout menopause, in particular, in the early-menopausal period. Thirdly, the number of WWH experiencing menopause is rapidly increasing, and there is an urgent need to conduct clinical studies designed to investigate bone disease throughout menopause, as older women living with HIV are generally rarely represented in HIV cohorts.

The limitations are intrinsic to the observational nature of this study. The lack of a control group was mitigated by the high number of repeated measures in the same patients, which allowed each patient to be the control of herself. Due to the late introduction of TAF into HIV care, we were not able to assess BMD and TBS trends in WWH transitioning to menopause. Additionally, the number of WWH with available TBS in both pre- and post-menopause was small, as the TBS tool became the standard of care at MHMC in 2019, giving us the possibility to only retrospectively analyze data from the past five years.

In conclusion, our results suggest that bone health should be assessed with BMD and TBS simultaneously, as they were associated with different risk factors. Both these biomarkers significantly decreased over time throughout the menopausal periods. The slope of the changes in BMD and TBS significantly decreased in the menopausal transition, suggesting that this period should be considered by clinicians as a key time during which to assess bone health and modifiable risk factors in WWH.

## Figures and Tables

**Figure 1 viruses-15-02375-f001:**
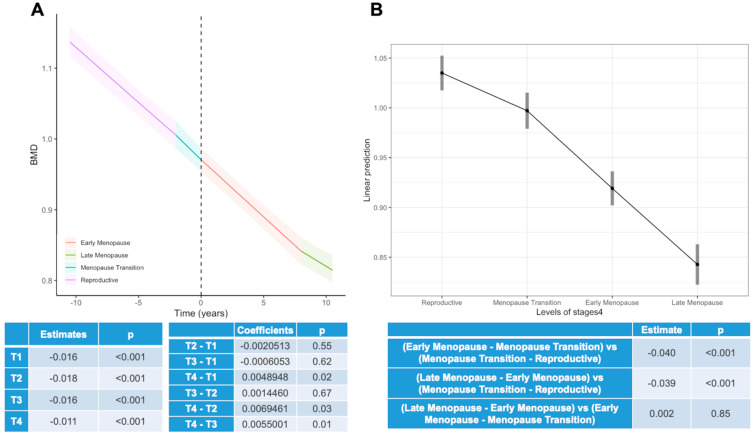
Trajectory of BMD throughout menopause. Panel (**A**) presents the slope coefficient of BMD change in the 4 STRAW+10 intervals. Panel (**B**) describes the average values of BMD in each transition period. Comparison derived from the STRAW+10 periods: (i) reproductive period, (ii) menopausal transition, (iii) early menopause, and (iv) late menopause.

**Figure 2 viruses-15-02375-f002:**
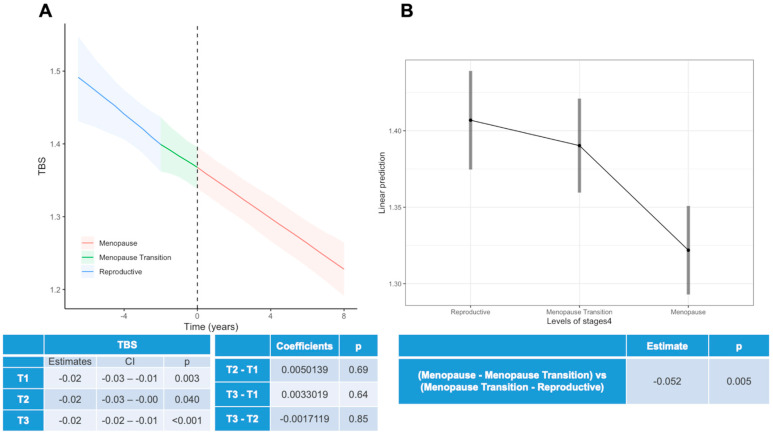
Trajectory of TBS throughout menopause, with panel (**A**) presenting the slope coefficient of TBS change in the 4 STRAW+10 intervals and panel (**B**) describing the average values of TBS in each transition period. Comparison derived from the STRAW+10 periods: (i) reproductive period, (ii) menopausal transition, and (iii) menopause.

**Figure 3 viruses-15-02375-f003:**
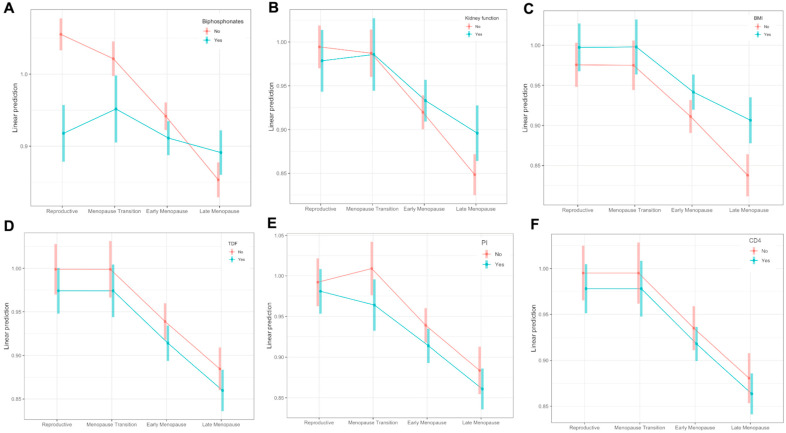
BMD trends throughout menopausal periods according to different risk factors: use of bisphosphonates (**A**), kidney function (**B**), BMI (**C**), TDF (**D**), PIs (**E**), and current CD4 cell count (**F**). Comparison derived from the STRAW+10 periods: (i) reproductive period, (ii) menopausal transition, (iii) early menopause, and (iv) late menopause.

**Figure 4 viruses-15-02375-f004:**
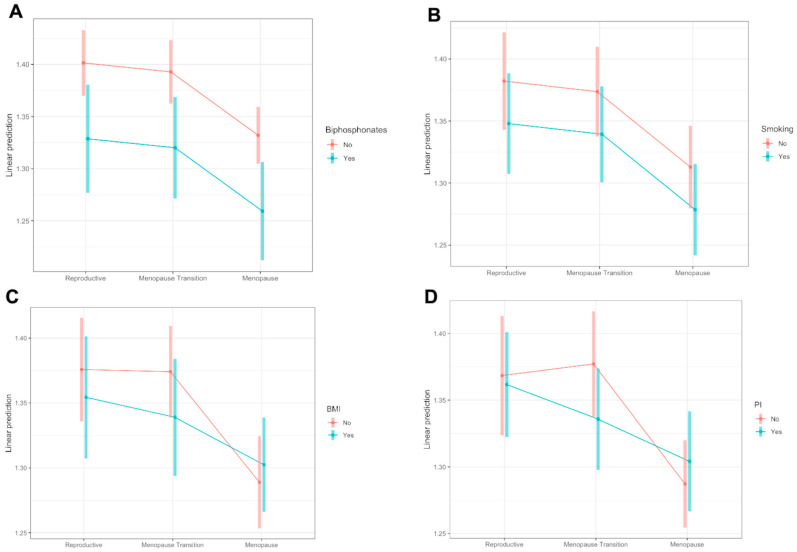
TBS trends throughout menopausal periods according to different risk factors: use of bisphosphonates (**A**), smoking status (**B**), BMI (**C**), and use of PIs (**D**). Comparison derived from the STRAW+10 periods: (i) reproductive period, (ii) menopausal transition, and (iii) menopause.

**Table 1 viruses-15-02375-t001:** Demographic, anthropometric, HIV, endocrinological, and DEXA characteristics according to menopausal status.

Characteristic	BMD			TBS		
	Pre-Menopause, N = 199	Post-Menopause, N = 199	*p*	Pre-Menopause, N = 48	Post-Menopause, N = 48	*p*
Demographic, anthropometric, and lifestyle characteristics						
Age, years, median (Q1, Q3)	46.2 (44.2, 48.4)	53.1 (50.7, 55.9)	<0.001 *	49.5 (47.2, 51.9)	53.5 (51.6, 56.2)	<0.001 *
Body mass index, kg/m^2^, median (Q1, Q3)	22.1 (20.2, 24.3)	22.5 (20.4, 25.2)	<0.001 *	22.4 (20.1, 25.7)	22.9 (20.4, 25.6)	0.001 *
Body mass index > 25 kg/m^2^, N (%)	54 (27%)	72 (36%)	0.003 *	14 (29%)	18 (38%)	0.20
Waist circumference, cm, median (Q1, Q3)	82 (77, 88)	84 (77, 91)	<0.001 *	80 (73, 90)	85 (78, 90)	<0.001 *
Currently smoking, N (%)	87 (44%)	80 (40%)	0.20	19 (40%)	17 (35%)	0.70
Alcohol use, N (%)	94 (47%)	88 (44%)	0.50	15 (31%)	16 (33%)	0.90
HIV-related variables						
Nadir CD4 cell count < 200 c/microL, N (%)	112 (56%)	115 (58%)	0.20	22 (46%)	22 (46%)	NA
Nadir CD4 cell count, c/microL, median (Q1, Q3)	183 (100, 254)	190 (108, 262)	NA	225 (138, 300)	225 (138, 300)	NA
Current CD4 cell count > 500 c/microL, N (%)	158 (79%)	177 (89%)	0.001 *	43 (90%)	43 (90%)	0.90
Current CD4 cell count, c/microL, median (Q1, Q3)	606 (433, 775)	766 (598, 930)	<0.001 *	712 (559, 889)	787 (568, 1074)	0.004 *
Years since HIV diagnosis, years, mean (SD)	18 (6)	25 (6)	<0.001 *	24 (5)	28 (6)	<0.001 *
Undetectable HIV RNA viral load, N (%)	173 (87%)	198 (99%)	<0.001 *	48 (100%)	47 (98%)	NA
ARTs						
Exposure to TAF, N (%)	2 (1.0%)	45 (23%)	<0.001 *	1 (2.1%)	12 (25%)	0.003 *
Exposure to TDF, N (%)	145 (73%)	96 (48%)	<0.001 *	25 (52%)	11 (23%)	<0.001 *
Exposure to protease inhibitors, N (%)	123 (62%)	104 (52%)	0.009 *	28 (58%)	19 (40%)	0.02 *
DEXA variables						
BMD, g/cm^2^, mean (SD)	1.03 (0.14)	0.92 (0.12)	<0.001 *			
TBS, median (Q1, Q3)				1.41 (1.35, 1.45)	1.32 (1.28, 1.39)	<0.001 *
FRAX, %, median (Q1, Q3)	2.96 (2.35, 4.21)	3.97 (3.09, 5.04)	<0.001 *	3.55 (2.95, 4.61)	4.11 (3.13, 4.84)	0.004 *
Endocrinological variables						
FSH, mIU/mL, median (Q1, Q3)	13 (8, 36)	70 (50, 94)	0.004 *	19 (7, 34)	77 (55, 97)	<0.001 *
LH, mIU/mL, median (Q1, Q3)	15 (7, 25)	35 (30, 40)	0.03 *	19 (6, 24)	41 (31, 48)	0.004 *
PTH, pg/mL, median (Q1, Q3)	53 (37, 66)	39 (29, 54)	<0.001 *	36 (25, 53)	33 (28, 48)	0.30
Phosphorus, mg/dL, mean (SD)	3.31 (0.49)	3.37 (0.38)	0.11	2.98 (0.43)	3.17 (0.43)	0.02
Estradiol, pg/mL, median (Q1, Q3)	87 (36, 154)	12 (7, 19)	0.004 *	40 (4, 15)	12 (5, 15)	0.90
Osteopenia and osteoporosis						
Osteopenia, N (%)	126 (63%)	151 (76%)	<0.001 *	29 (60%)	31 (65%)	0.50
Osteoporosis, N (%)	31 (16%)	71 (36%)	<0.001 *	10 (21%)	12 (25%)	0.50
Use of bisphosphonates, N (%)	7 (3.5%)	32 (16%)	<0.001 *	4 (8.3%)	5 (10%)	0.90

* *p* < 0.05.

## Data Availability

The datasets generated during and/or analyzed during the current study are not publicly available, but are available from the corresponding author on reasonable request.

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
