# Peer review of "Bone Mineral Density and Trabecular Bone Score Changes throughout Menopause in Women with HIV"

_viruses, 2023, doi:10.3390/v15122375_

Round 1

Reviewer 1 Report

Comments and Suggestions for Authors

This retrospective longitudinal study compared vertebral BMD and TBS for bone loss prediction in women with HIV (WWH), across the menopausal transition. Bone loss due to HIV, AIDS, and ART complications is established in WWH, and elevated fracture incidence is well documented. Little is known about the effects of menopause on bone condition in WWH L, making this study timely and important as the average age of PWH is now ~50 yoa with average onset of menopause ~51 yoa. Furthermore, DEXA based BMD assessments have significant shortcomings and supplemental methodologies such as TBS, that can better assess architectural properties of trabecular bone in the vertebrae, have been developed to improve fracture prediction.

Although in some studies TBS has been found to be more sensitive that DEXA alone, in this study while DEXA demonstrated a significant decline in BMD, TBS changes were less informative and failed to achieve statistical significance in most case suggesting that TBS may not be suitable for augmenting DEXA in assessing postmenopausal changes in WWH.  Overall, the studies are well performed and the manuscript is clearly written, and the data will be of interest to the HIV-bone research community.  

Major comment:

A limitation of this study is the lack of an HIV negative control group. It is well-established that BMD declines after the menopause in HIV negative women, thus it is expected that this would happen in WWH too. An important question in the field is whether postmenopausal bone loss is exacerbated in the context of HIV leading to more osteopenia/osteoporosis. It is thus surprising that the authors neglected to examine BMD-derived Z-scores, which are automatically generated by the DEXA software. This would have provided a reference/normalization against an age-matched healthy control group and allowed an estimate of the cumulative effects of HIV/ART and menopause on BMD and osteopenia/osteoporosis in WWH. A Z-score analysis is recommended and could significantly increase the impact of the work.

Other Comments:

1.     Introduction: Although TAF does slightly lessen the degree of bone loss associated with TDF, the notion that TDF and PIs promote bone loss just through “bone toxicity” is a dated concept and numerous studies show that the dominant effects of ART on bone are likely independent of drug type (Brown et al. J. Acquir. Immune Defic. Syndr. 2009;51(5):554-561. (PMID 19512937) and the aggressive ART-induced bone loss common in severely immunocompromised people, is mediated through inflammatory responses related to CD4 rebound (McGinty et al Curr. Opin. HIV AIDS. 2016;11(3):253-260. (PMID 27008474). Consistent, with this notion, only relatively small (direct) effects of TDF on bone are observed when TDF is used prophylactically (PrEP) in HIV negative people (Mulligan et al Clin. Infect. Dis. 2015;61(4):572-580. (PMID 25908682)), or when TDF is used to treat HBV. The authors should revise the introduction to provide a more balanced overview of ART action on bone, to better inform non-expect readers of the significant and often underappreciated problem that ART still poses.

2.     This multimodal action of ART should also be considered when interpreting the sub-analysis related to TDF and TAF.

3.     The manuscript refers throughout to use of the STRAW menopausal staging system, but in fact, the updated 2011 revised “STRAW+10” system was used! This is a significantly upgraded system integrating hormonal levels to augment staging. The authors should thus change STRAW to STRAW+10 throughout, to avoid reader confusion.

4.     In some places the authors use “DXA” abbreviation and in some (most) “DEXA”. Please select one abbreviation and use consistently throughout.

5.     Lines 169-171: It is important to emphasize for the non-expert reader, that TBS generates a weak extrapolation of microarchitecture from a low resolution 2D technique, while µCT is a high-resolution 3D technique. TBS does not provide anything even close to approximating the detailed structural indices that can be achieved by CT-based analyses. In fact, µCT is not a technique commonly applied to human studies, and HR-pQCT or pQCT are probably a more appropriate techniques for use as comparisons.

6.     Line 244, anti-absorptive should be “anti-resorptive”.

7.     Line 377-379, the discussion on CD4 cell count focuses exclusively on CD4 changes with age, however the studies followed WWH over only a median of 12 years. Likely more relevant are the significant inflammatory effects associated with estrogen deficiency on CD4 T cells. Other significant contributors to bone loss in HIV/ART are reported to be CD4 number prior to ART, and the magnitude of CD4 expansion after ART is shown to correlate with the magnitude of bone loss.

Comments on the Quality of English Language

Overall, the manuscript was well-written and easy to understand. There are some minor grammatical issues throughout that would require a Native English speaker to detect and correct.

Author Response

Dear Editor,

We are very grateful for the acceptance of our paper entitled: “Bone mineral density and trabecular bone score changes across menopause in women with HIV”.

We provided a point-by-point reply to the comments raised by the reviewers, and we incorporated the related changes in the manuscript.

Reviewer 2 Report

Comments and Suggestions for Authors

The objective of the study of Milic Jovana et al. was to describe
trajectories of bone quantitative 26 tity (BMD) and bone quality (TBS) changes across pre-menopause and post-menopause, in WWH 28 undergoing different ART and explore risk factors associated with those changes.
The article is well written, and the statistical analysis is adequate,
even if there are some points to improve. My comments are below.
1.         I suggest to the authors specify in the Statistical analyses
section, the normality test was used.
2.         In addition, authors should add an asterisk next to significant
p-values or write them in bold in the tables, to improve their
readability.
3.         Authors should insert a limitations section below the Discussion section
4.         In the manuscript, the authors should not use the form " we
demonstrated", since the results are based on statistical analysis,
i.e. in the probability theory, it is preferable to use the form " we
observed " or " we found"
5.         The reference section shows about 75% of references more than ten
years old. Authors should add more recent references and discuss them
6.         The Sample size estimation section is missing.

Comments on the Quality of English Language

The manuscript might need moderate language editing.

Author Response

(The authors gave the same response as above.)

Reviewer 3 Report

Comments and Suggestions for Authors

Comments to the Authors of manuscript number: viruses-2423227 entitled “Bone mineral density and trabecular bone score changes across menopause in women with HIV”.

1. L 26 – TBS – please explain

2. abstract: - due to many abbreviations it is to hard understand

3. L 65- what is WWH?

4. L 65 – it is not understand at all

5. L 95-99 – the lack of hypothesis

6. L105 – NCDs?

7. L 104, 106 – if ovarian assessment, people or females?

8. L 122 – MHMC?

9. L 127 –recommended by whom?

10. L 148 – TAF?

11. L 149 – INSTI?

12. L 154 – EACS?

13. L 160 – describe please the apparatus

14. L 165 – producer

15. L 177 – producer

16. L180 – OO-?

17. L 202- for p?

18. L 203 – version of software, producer

19. the lack of the description of HIV related variables in Material and methods

20. the lack of the description of endocrinological variables in material and methods. Were performed by ELISA? It should be described in details.

Author Response

(The authors gave the same response as above.)

Round 2

Reviewer 1 Report

Comments and Suggestions for Authors

My major comment has been satisfactorily addressed, as well as most of the minor comments.

One remaining minor comment should be addressed because as it currently stands it is incorrect and misleading.

Comment # 5 and Page 4:

My comments regarding microCT were not understood by the authors. In contrast to what is stated in the manuscript, uCT, although having very high resolution (typically a voxel size of ~6um) is exclusively a preclinical technology that is only suitable for very small bones such as mice and rats in vivo and bones of some other small mammals ex vivo. It is not compatible with human applications, especially in vivo. The comparable technology used in humans is called high-resolution peripheral quantitative computed tomography (HR-pQCT) with a voxel resolution of ~ 60 um. Some studies have also used clinical computed tomography (CT) scanners which have a lower resolution than HR-pQCT (~350 um).

Comments on the Quality of English Language

Overall the English is good. There are a few typos and grammatical errors typical of most manuscripts.

Author Response

Reviewer 1

My major comment has been satisfactorily addressed, as well as most of the minor comments.

One remaining minor comment should be addressed because as it currently stands it is incorrect and misleading.

Comment # 5 and Page 4:

My comments regarding microCT were not understood by the authors. In contrast to what is stated in the manuscript, uCT, although having very high resolution (typically a voxel size of ~6um) is exclusively a preclinical technology that is only suitable for very small bones such as mice and rats in vivo and bones of some other small mammals ex vivo. It is not compatible with human applications, especially in vivo. The comparable technology used in humans is called high-resolution peripheral quantitative computed tomography (HR-pQCT) with a voxel resolution of ~ 60 um. Some studies have also used clinical computed tomography (CT) scanners which have a lower resolution than HR-pQCT (~350 um).

Authors’ answer:

We thank the reviewer for the comment, which helped us to further clarify this section, that now reads as follows:

“TBS is a recently introduced metric that analyses gray-level textual information in the lumbar spine DEXA image to approximate trabecular microarchitecture. TBS generates extrapolation from a low resolution 2D technique and it is not impacted by bone size.

TBS values increase with enhanced skeletal texture (i.e., better microarchitecture) and decrease with deteriorated skeletal texture (i.e., poorer micro-architecture). Numerous studies found a correlation between TBS texture parameters and microstructural parameters of bone determined via micro-CT (exclusively preclinical technology and not suitable for humans), indicating a relationship between TBS and 3D microarchitecture [31,32].”

Reviewer 3 Report

Comments and Suggestions for Authors

I have no more comments

Author Response

We thank the reviewer for the comments that helped us to significantly improve the manuscript.